# Spatial dependence in the rank-size distribution of cities – weak but not negligible

**Rolf Bergs** *

PRAC, Bad Soden, Germany

* RolfB@prac.de

**Data Availability Statement:** Data are available from Harvard Dataverse: https://dataverse.harvard.edu/dataset.xhtml?persistentId=doi:10.7910/DVN/EK4CNU

**Funding:** The author is affiliated as partner with PRAC. PRAC is a private institute organised as a

## Abstract

Power law distributions characterise several natural and social phenomena. Zipf's law for cities is one of those. The study views the question of whether that global regularity is independent of different spatial distributions of cities. For that purpose, a typical Zipfian rank-size distribution of cities is generated with random numbers. This distribution is then cast into two different settings of spatial coordinates. For the estimation, the variables rank and size are supplemented by considerations of spatial dependence within a spatial econometric approach. Results suggest that distance potentially matters. This finding is further corroborated by four country analyses even though estimates reveal only modest effects.

## 1 Introduction

Zipf's law of the rank-size distribution of cities is regarded as an enthrallment of rare social physics. Krugman [1] has described this phenomenon even as an embarrassment for economic theory (p. 42–46). At first glance, the relationship between size and rank suggests a tautological relationship because size directly determines rank and vice versa. The independent variable is thus perhaps no true predictor but could just be part of a simple universal statistical phenomenon. The power law exponent typically close to -1 and a determination coefficient ($R^2$) close to one are an indication of that. Therefore some emphatically questioned the relevance for economic analysis, notably Gan *et al.* [2]. Yet, such a striking ubiquitous regularity has motivated the exploration of hidden explanatory factors behind. Various authors, such as Gabaix, Fujita *et al.*, Brakman *et al.*, Reggiani and Nijkamp and Ioannides have done this [3–7]. In contrast to the Zipf distribution of frequency of words in languages [8], the rank-size distribution of cities appears slightly more varied between countries as e.g. shown by Rosen and Resnick [9] and less stable over time as e.g. found by Brakman *et al.* [10] but there is a secular convergence that is explained by Gibrat's law and its resulting steady state [3]. The fact that this happens in all countries, regardless of their economic structures and histories, still lacks a truly sufficient explanation. When comparing such power law distributions for different types of data one criterion could perhaps add minor insight: A spatial versus non-spatial context. Spatial dependence in terms of contiguity or distance between cities of different rank or size may affect Zipf's law for cities; in contrast, space can never predict the rank-distribution of words.

partnership company. The research represented by the paper at hand is a secondary outcome of a current research project, being funded by the European Commission „Horizon2020" programme (grant No. 727988). The funder has not been involved in the study design, data collection, decision to publish and preparation of the manuscript. The only criterion to be fulfilled by the author is the thematical relevance of this manuscript for the purpose of the a.m. research project. The funder provided support in the form of salaries for authors [RB], but did not have any additional role in the study design, data collection and analysis, decision to publish, or preparation of the manuscript. The specific roles of these authors are articulated in the 'author contributions' section. My commercial affiliation (PRAC) did not play any role in this context. The paper was exclusively prepared and written by myself.

**Competing interests:** The author is affiliated with PRAC. This does not alter our adherence to PLOS ONE policies on sharing data and materials.

To have a closer look at that context, I first reflect on theoretical considerations of spatial dependence in Zipf's law before simulating a typical rank-size distribution of cities for a varied distribution of spatial coordinates. The objective is to see how much influence spatial distance could have on the ranks of cities and thus on the shape of the distribution. Especially in countries with a geographical concentration of bigger cities there is reason to assume that these cities have evolved due to certain spatial advantages (e.g. raw materials, climate, accessibility or certain random determinants). Those city clusters are often characterised by specific industries of national importance. Whether and how dispersion and concentration forces determine the rank-size distribution of cities has been a widely researched object in urban economics.

Surprisingly, there has been little research shedding light on spatial dependence associated with Zipf's law. Lalanne views the dichotomic urban structure of Canada [11]. She rejects the Zipf law and its underlying scale invariance and shows that the Canadian urban structure has evolved in a deterministic process based on urban size (inhabitants within administratively defined boundaries), previous growth and the spatial setting. Coefficients for the years 1971 to 2001 vary between -0.77 and -0.81. The spatial component is not part of the Pareto regression; instead growth of cities is regressed on size and previous growth in standard spatial regression models (SEM and SAR). Le Gallo and Chasco explore Zipf's law for Spain by applying a SUR model which they cast into spatial autoregressive and error specifications. Zipf's law does not hold between 1900 and 2001. While the simple OLS estimate varies between -0.54 and -0.66, the extended spatial models deviate even further from Zipf's law, thus revealing spatial impacts [12]. Cheng and Zhuang [13], who look at urban evolution in central China under consideration of Zipf's law, show that the estimation of the Pareto coefficient has displayed an undulatory pattern between 1985 and 2009. They cannot confirm Zipf's law at any point of time. The OLS estimates are then augmented by the use of spatial autoregressive or spatial error specifications. Like in the study of Le Gallo and Chasco [12], spatial dependence increases the deviation from Zipf's law. It is, however, to be stressed that in the three studies on Canada, Spain and China the size of cities is not defined by functionality but the number of inhabitants within administratively defined boundaries of all cities, i.e. including the lower tail of the distribution. This is a reason why Zipf's law often does not hold [5: 301–306].

In the study at hand I intend to show (i) whether and how spatial dependence of a Zipfian rank-size distribution varies among different geographical settings of cities and (ii) how these settings behave differently along the entire distribution of cities and specifically the upper Pareto tail. Evidence suggests that the rank-size distribution is not homogeneously following a Pareto shape, but rather a combination of an upper Pareto and a lower lognormal section. By using a switching model, Ioannides and Skouras [14] show for US cities with 2000 Census places data that there is a narrow transition corridor around slightly more than 60,000 inhabitants where the upper tail Pareto distribution merges with a lower tail lognormal distribution. They reject Eeckhout's standpoint [15], that the entire rank-size distribution of cities is best described by a lognormal distribution. Evidence of the hybrid distribution was corroborated by several further studies, e.g. Malevergne *et al.* [16], so that this is explicitly considered in my paper.

In this paper I first explore an ideal type random-generated hybrid distribution with two different spatial settings. This should demonstrate that the chosen econometric methodology is powerful enough to detect distance effects if the patterns are sufficiently strong. The simulation exercise is then followed by four country studies covering the USA, the United Kingdom, Germany and Slovenia, the latter representing a former smaller province of Yugoslavia. For Slovenia, not only population was used as the size variable but also the detected extent of natural cities to better represent their true functional size. In essence, the paper is a theory-led artificial simulation of Zipf's law enriched with real world studies.

## 2 Some theoretical considerations

The spatial relevance of the rank-size distribution of cities was already emphasised by Zipf himself in his widely noticed lemma [17]: If effort of interaction among the possible pairs of cities is optimum (with least effort for all individuals), the cities (settlements) of different size are ranked in a way so that the total population of a country $S_c$ equals the sum of a harmonic series:

$$S_c = \frac{S_p}{1^\alpha} + \frac{S_p}{2^\alpha} + \frac{S_p}{3^\alpha} + \cdots \frac{S_p}{r_n^\alpha} = \sum_{r=1}^{n} \frac{S_p}{r^\alpha}, \tag{1}$$

where $S_p$ is the population of the primacy city, $r$ is the rank of an individual city and $\alpha$ is the power exponent determining the shape; in case of a perfect Zipfian relationship, the cumulative distribution function then follows

$$S = BR^{-\alpha} \tag{2}$$

or in its reversed Pareto form with $R$ (rank) as the dependent variable:

$$R = CS^{-\alpha} \tag{3}$$

with $\alpha = 1$ and $B = C$. This power law describes a scale-invariant pattern with very few large and very many small items as it is found in many natural systems. According to Zipf, the slope of that particular distribution necessitates the effort of interaction between the communities to be at a minimum when multiplied by the distances $d$ between the communities. Zipf's lemma describes a stylized equilibrium model in which there is a scattered distribution of settlements close to the raw materials (first economy) and one big city where all the raw materials are processed (second economy). Since living in either place will create opportunity costs for any dweller, both economies are in conflict over unification and diversification. The conflict between those forces plays a central role in the determination of the effort-minimizing number, location and sizes of settlements or, with other words: the built environment is created so that costs of primary production, processing, and the transport of goods and factors between the two economies are minimized. Obviously, an equilibrium is found when the magnitude of the centrifugal and centripetal forces is equal. In this optimum case $\alpha = 1$, and the equilibrium is then Pareto efficient. If one imagines a growing network of cities, it becomes obvious that the number of connections represents an economic value (utility), and this again is highest the minimum possible effort is needed. As explained by Kak [18], the value of a potential network with $n$ items (cities) then grows in proportion to $n \cdot \log(n)$. This explains a power law behavior as a precondition of least effort.

According to Zipf [17], this pattern only works in social systems that exactly produce what they consume and where all members of the population receive an equal share of the national income. This understanding very obviously assumes constant returns to scale in both economies. Consequently, if the system is not in an equilibrium (e.g. with the occurrence of increasing returns), diversifying (centrifugal) and unifying (centripetal) forces do not offset each other. In this case the slope of the power law changes. It is to be stressed that Zipf's spatial equilibrium essentially depends on the existence of spatial heterogeneity. Perfect divisibility of space would rule out any equilibrium (Starret's spatial impossibility theorem) [5, 19]. Hence, (i) for cities to evolve anyway, indivisibility is needed and (ii) for cities to evolve efficiently with optimum allocation of resources needed for interaction, their rank-size distribution should converge to Zipf's law. This deserves some closer examination since indivisibility of space unveils an important explanatory limitation of Zipf's considerations of a spatial equilibrium.

In Zipf's model, the difference between the first and second economy is solely explained by their respective functional roles. However, the evolution of the spatial economy, characterised by a dynamic rural-urban differentiation, essentially exhibits spatial factor and goods price differentials that originate from several interrelated determinants, such as increasing returns in manufacturing production [20], monopolistic competition and higher real urban income through the supply of a variety of substitutable goods and lower transport costs [21], stronger knowledge spillovers in agglomerated urban settings [22], trickle-down effects of individual specialised skills on the local qualification and productivity levels [23] or agglomeration fuelled by entrepreneurial uncertainty and risk [24] to mention some. In those settings consumers aim to maximise their utility not only by minimising effort of transport but also to maximise their real income through an optimum choice of expenditure on food and on the variety of substitutable manufactured products. The "love for variety effect", scale economies and less transportation effort are circularly caused. They are a bonus for larger markets [5] and constitute an urban amenity.

Interestingly, the major thread of the subsequent theoretical literature on Zipf's law since the 1950s centered around statistical and largely non-economic explanations based on random growth of population [25]. Later the size- and variance-independent growth of cities was discussed to explain the inherent fractal dimension of Zipf's law by Gibrat's law. In these models the potential spatial dimension was largely ignored. Indeed, in Gabaix's [3] model of zero normalized city growth, space and distance between cities do not suggest to be meaningful factors under the strict assumption of Gibrat's law. In this model, the economic foundation of Gibrat's law is explained by scale-independent regional and policy shocks with the same variance for all cities in addition to specific shocks that affect particular industries, thus implying a decreasing variance with city size. However, for the upper tail of the city size distribution, industrial shocks may die out so that, according to Gabaix, variance rather depends on the policy and regional shocks.

While urban economics and regional science in the 20th century generated an abundance of theoretical models to explain agglomeration economies, most of them had pursued a partial focus. During the last twenty years, economic geography models became more consolidated as to nest different hypotheses that may constitute forces of agglomeration. An important seed of those efforts was the seminal work of Fujita, Krugman and Venables [4] which reveals the sensitivity of a spatial evolution path simulated by the relationship between the share of manufacturing employment and transport costs adjusted with few decisive parameter settings (substitution elasticity, *iceberg* losses). Those determine tipping points (bifurcations) causing either centrifugal or centripetal spatial evolution paths. This work also contributed to more insight into the economic determinants of the rank-size distribution of cities. Brakman et al. [5] develop a core economic geography model with monopolistic competition and further extensions to explore the behavior of the spatial economy under different parameter adjustments. By extending their model with congestion, as the major counterforce of agglomeration, they simulate Zipf's law historically and show that there is an N-shaped pattern of the Zipf coefficient over time from the pre-industrial to the post-industrial era (for log of city size as the dependent variable and log of rank as the predictor). In this model, aimed to explain Zipf's law, space is explicitly considered, but in terms of agglomeration, rather than inter-city distances.

Only recently, a growing record of research stressing the relevance of distance and accessibility in the evolution of urban space can be observed. Indeed, the functional differentiation between Zipf's first and second (spatial) economy closely corresponds to the Central Places theory provided the existence of agglomeration economies is not ignored. Distance, or the effort to cover it, is then the major friction in city interaction. This makes spatial distance not

only important for city interaction but eventually also relevant to the rank-size distribution of those cities.

In a highly comprehensive analysis to capture the determinants of Zipf's law economically and spatially, Ioannides [7] demonstrates the limits of explanatory power of otherwise well-founded theories. This relates to independently and identically distributed (i.i.d.) growth rates of cities when using normalised city sizes and to an entirely lognormal city size distribution determined by Gibrat's law [3, 15]. In a growing and increasingly urbanizing economy the assumption that city growth rates are essentially i.i.d. may not further hold. Especially the relationship between the variation of fixed costs and the number of production sites appears to be an important element to explain a power law distribution in the upper tail of the city sizes. The finding reveals important explanatory power of the Central Places theory. Firms with lower fixed costs are spatially more dispersed (i.e. in big and small cities), while those with high fixed costs locate close to those with lower fixed costs (usually in larger cities). This refers to the work of Hsu [26] who concludes that ". . . The power law for cities and firms and the NAS ["Number-Average-Size"] rule arise when the distribution of scale economies is regularly varying. In fact, this is the condition for ensuring that a central place hierarchy is a fractal structure. . ." (p. 923). The Central Place theory can be thus reconciled with the mainstream economic theories and a power law behavior of the upper tail of the city distribution: Larger cities are not only more diversified than small ones because many small and big industries are agglomerated there but because bigger cities specialize in industries with higher scale economies.

In a very recent analysis of this thread of studies, Mori *et al*. [27] compare real with random city systems at national level and within the hierarchy of a system with central places. They find strong evidence of a fractal dimension in the rank-size distribution of cities, but this is not governed by random growth of cities but rather by local city systems surrounding major cities. In another study, Jiang *et al*. [28] explores the system of cities from the viewpoint of the design of space and finds that cities are not isolated but coherent entities within a connected whole, whereas cities themselves comprise coherent hotspots. Also arguing with the Central Place theory, Jiang concludes that the order of the built environment corresponds to the order in nature and that scaling law and spatial dependence ". . . are fundamental not only to geographical phenomena, but also to any other living structure that recurs between the Planck length and the size of the universe itself. . . ." (p. 311).

In addition to economic geography models there are also further recent contributions aimed to explain a spatial Zipf law from the viewpoint of social physics. With a probability based approach of urban evolution, Rybski *et al*. simulate formation and growth of cities under the assumption of Tobler's law, namely that urban growth takes place close to other urban settlements [29]. In comparative simulations with different numbers of iterations and different strengths of distance decay $\gamma$ they show, that for sites in a grid with a central site already occupied ($w = 1$) any other site $j$ in that grid ($w = 0$) will be occupied with a probability:

$$p_j = A \frac{\sum_{k \neq j} w_k d_{j,k}^{-\gamma}}{\sum_{k \neq j} d_{j,k}^{-\gamma}}, \tag{4}$$

where $d_{j,k}$ is the Euclidian distance between locationas $j$ and $k$, and $A$ is a normalized constant so that the maximum probability is 1. Hence, in this simulation, supported by a real world study on the urban development of Paris, evolution of new sites solely depends on distance to sites already existing. Despite the fact that the simulated urban evolution is not random but deterministic the authors confirm Zipf's law and scale invariance of the clusters generated,

except the primate one. Thus, Zipf's law can be also reproduced by "spatial explicit preferential attachment".

A well-known example of such peculiar spatial trajectories is the Ruhr area in Germany. Here the theory of Central Places seems to fail in explaining the fact that bigger cities (more than 100,000 inhabitants) are just medium centers or even cities with minor central relevance. Findings of Dobkins and Ioannides [30] for US cities also confirm that large cities tend to have large neighbors. This may suggest a possible inconsistency with the Central Place theory, not for the regional setting of cities (as new neighbors entering are still relatively small compared to the older ones) but perhaps from a national viewpoint with a larger variation of city sizes within the different centrality classes. Here it may happen that size of cities in urban clusters does not not anymore correspond to centrality. But also an opposite type of spatial settings is imaginable, such as for big urban areas surrounded by only particularly small municipalities with little centrality function, e.g. the Berlin urban zone.

Such peculiar agglomeration phenomena then essentially imply positive or negative spatial autocorrelation of city ranks or sizes and may have an influence on the Pareto coefficient at national scale, so that size and growth of such cities are not necessarily random but spatially autoregressive (or disturbed by spatially autocorrelated error) and thus partly depending on rank or size of neighbor settlements and their growth.

In the real world, Zipf's law for cities is never absolutely perfect. Many empirical studies have shown this, as mentioned earlier. Reasons for that can stem from the regional political economy, notably a functionally inadequate administrative delineation of urban space or the politically emphasized weight of the primate city (in many cases the capital). A further reason can be the typical hybrid distribution form mentioned earlier (Pareto and lognormal) and whether the full or a curtailed range of city sizes is regarded; coefficients estimated can differ strongly. In addition to such data issues, increasing returns or congestion can have an influence on the rank-size distribution of cities. This does not at all mean that Zipf's law fails in such cases. A spatial distance influence can improve the fit of the power law or it can move $\alpha$ further away from 1. But as long as evolution of big cities in urban clusters tends to exhibit spatial dependence in the rank-size distribution, its effects would be essentially concealed by a non-spatial regression. In conclusion, this suggests a spatial econometric approach when testing Zipf's law.

## 3 Methodological approach

The central assumption is as follows: For the typical upper Pareto (3,1) tail the expected exponent $\alpha$ is approximately 1 ($\pm 0.1$) as empirically confirmed ubiquitously. In its log-linear form the above described cumulative distribution function (3) to be estimated is:

$$\ln(R) = \ln(C) - \alpha \ln(S) + \varepsilon \qquad (5)$$

where the residual error is assumed i.i.d with $\varepsilon \sim N(0,\sigma^2)$. OLS or maximum likelihood are possible estimators of $\alpha$. For the combined Pareto and lognormal tails the exponent usually does not fit Zipf's law but only for the Pareto section. In addition to that, the theoretical considerations put forward earlier suggest that the rank-size distribution of cities is potentially affected by spatial distance in the sense of Tobler's law. Such spatial forces are however not incorporated into Zipf's law, so that in theory a Pareto exponent $\alpha \approx 1$ in one country may remain stable under consideration of spatial dependence of rank while, in another one, the incorporation of such interaction could perhaps lead to a minor or major change of $\alpha$. For spatial dependence to be considered for the rank-size distribution of cities, some methodological considerations are relevant: Compared to gravity estimations that address the interaction of

places (i.e. the number of combinations), the estimation of the rank-size distribution cannot include distance as one regular predictor. Either distance enters the model as a large matrix of single independent variables (one for every city combination) or one controls for spatial spillover or error of the residuals in regression analyses. It is to be stressed that a stand-alone construction of numerous independent distance variables would ignore the possible endogeneity of distance (spillover effects of the dependent variable or residual spatial autocorrelation). The underlying economic rationale is the utility of interaction with respect to spatial distance between cities of either similar or very different ranks. A more precise approach would be thus a spatial econometric procedure [31]:

$$\ln(R) = \rho W \ln(R) + \ln(C) - \alpha \ln(S) + \varepsilon \ (\textbf{SAR}) \tag{6}$$

or

$$\begin{cases} \ln(R) = \ln(C) - \alpha \ln(S) + v \ (\textbf{SEM}) \\ v = \lambda W v + \varepsilon \end{cases} \tag{7}$$

where $W$ is a N x N row-standardised weight matrix (inverse distance) to capture a potential distance effect and $C$ is a constant while $\rho$ (spatial spillover) and $\lambda$ (spatial autocorrelation in the residuals) in addition to $\alpha$ (direct effects) are the coefficients estimated. The error term $v$ in the SEM case consists of spatial error and the residual $\varepsilon$.

The right choice between both models can be determined by different tests, such as the z-score of Moran's I of the residuals and (Robust) Lagrange multiplier (LM) tests. The different estimation types in Tables 1 and 2 correspond to the respective choice.

Spatial lags can be also expected for the independent variable. In an extended Spatial Durbin model, both, the dependent as well as independent variables appear simultaneously as lagged variables. As proposed by Halleck Vega and Elhorst, a simpler approach to consider the spatial lag of the predictor is offered by the SLX model [32]:

$$\ln(R) = \ln(C) - \alpha \ln(S) \pm \theta W ln(S) + \varepsilon. \tag{8}$$

The coefficients $\alpha$ and $\theta$ can be estimated by OLS. This model is applied in addition to the SEM/SAR estimations to control for spatial dependence of the variable $S$.

One major shortcoming of all such spatial econometric procedures needs to be stressed: Inverse distance never properly represents the effort needed to access a close or distant place. Natural transport infrastructure, topographical characteristics and the energy resources available also determine mutual accessibility and city interaction. Distance is thus only a proxy for effort and time needed, given it is understood in the same way as Zipf had defined the problem in his lemma. But this caveat applies to all such spatial econometric models as long as there are no differentiated data that can replace inverse distance in the spatial weight matrix. In the end, a comparative view over time (different years) based on a true effort-specific weight matrix could much better reveal the important dynamic of spatial dependence in specific geographical settings during phases of major structural change. But this would be a subject of future research.

## 4 Data

The two simulated "countries" describe a distribution with in each case 109 observations for cities, the upper 50 being Pareto (3,1) distributed and the lower 59 with a lognormal shape. Both distributions are consecutively random-generated and then matched into one data set. The first step is the generation of 100 observations for both distributions. The upper 50 observations of the Pareto set are then matched at the point with the next smaller observation in the lognormal

**Table 1. Simulated spatial extension of Zipf's law (full rank-size distribution).**

| Endogenous variable: ln(Rank) | I | II |
|---|---|---|
| ln(Size) | -0.403 | -0.322 |
| (Standard error) | (0,024)*** | (0,028)*** |
| Constant | 3.871 | 1.057 |
| (Standard error) | (0,037)*** | (2,998)*** |
| $\lambda$ | -0.321 | 0.969 |
| (Standard error) | (0,650) | (0,030)*** |
| $\rho$ | | |
| (Standard error) | | |
| Log Likelihood | -78.581 | -44.774 |
| Wald-Test $\lambda$ and $\rho = 0$ | | |
| $\chi2$ | 0.244 | 1016.678 |
| z-score: Moran's I (resid.) | 0.141 | 10.081 |
| (p) | (0,888) | (0,000) |
| Lagrange Multiplier (LM) | 0.163 | 80.748 |
| (p) | (0,686) | (0,000) |
| Robust LM | 0.028 | 15.462 |
| (p) | (0,867) | (0,000) |
| Obs. | 109 | 109 |
| OLS | | |
| ln(Size) | -0.403 | -0.403 |
| (Standard error) | (0,025)*** | (0,025)*** |
| Constant | 3.870 | 3.870 |
| (Standard error) | (0,049)*** | (0,049)*** |
| Adj. $R^2$ | 0.709 | 0.709 |

Note: Column I shows estimates for the arrangement of all city ranks with a normal distribution across space (see Fig 1) while column II displays the respective estimates for the geographically ranked arrangement of cities (see Fig 2). The choice between either SAR or SEM was determined by the z-score of Moran's I of the residuals and the Lagrange multiplier test statistic.

Source: Own data simulations.

set. The proportion of observations in both tails could be different, e.g. exhibiting a larger log-normal tail, but this would only affect the shape of the full distribution. The only purpose is to simulate one realistic rank-size distribution and to explore how sensitive it reacts on changing geographic coordinates. In the next step this hybrid rank-size distribution is combined with different distributions of coordinates X and Y, the first one being randomly generated to fit a normal distribution (Fig 1). Based on this configuration a spatial weight matrix is derived. With a normal distribution of both X and Y coordinates big and small cities are spread evenly.

In the second variation (Fig 2) with the same rank distribution the normally distributed coordinates of X and Y are both ranked as well, so that all cities are geographically positioned on a diagonal line, ordered along rank, the biggest city in the outer North-East, the smallest one in the outer South-West (like a one-dimensional von-Thunen assembly).

It is expected that in this case both, rank as well as size, exhibit spatial autocorrelation even though such a setting is hardly encountered in the real world.

The only purpose of this extreme setting is to show the possible potential of distance impact depending on the distribution of coordinates. With other words, identical coefficients confirming Zipf's law may have a different meaning for different countries.

**Table 2. Simulated spatial extension of Zipf's law (rank-size distribution for the Pareto tail).**

| Endogenous variable: ln(Rank) | I | II |
|---|---:|---:|
| ln(Size) | -0.995 | -0.862 |
| | (0,018)*** | (0,029)*** |
| Constant | 5.086 | 3.846 |
| | (0,430)*** | (0,208)*** |
| λ | - | - |
| | - | - |
| ρ | -0.056 | 0.262 |
| | (0,145) | (0,050)*** |
| Log Likelihood | 39.618 | 50.343 |
| Wald-Test λ and ρ = 0 | | |
| χ2 | 0.149 | 27.289 |
| z-score: Moran's I (resid.) | - | - |
| (p) | - | - |
| Lagrange Multiplier (LM) | 0.146 | 23.817 |
| | (0,703) | (0,000) |
| Robust LM | 0.081 | 14.067 |
| | (0,776) | (0,000) |
| Obs. | 50 | 50 |
| OLS | | |
| ln(Size) | -0.995 | -0.995 |
| | (0,018)*** | (0,018)*** |
| Constant | 4.921 | 4.921 |
| | (0,039)*** | (0,039)*** |
| Adj. R$^2$ | 0.984 | 0.984 |

Note: Column I shows the estimates for arrangement of the Pareto tail of the city ranks with a normal distribution across space (see Fig 1) while column II displays the respective estimates for the geographically ranked arrangement of cities (see Fig 2). The choice between either SAR or SEM was determined by the z-score of Moran's I of the residuals and the Lagrange multiplier test statistic.

Source: Own data simulations.

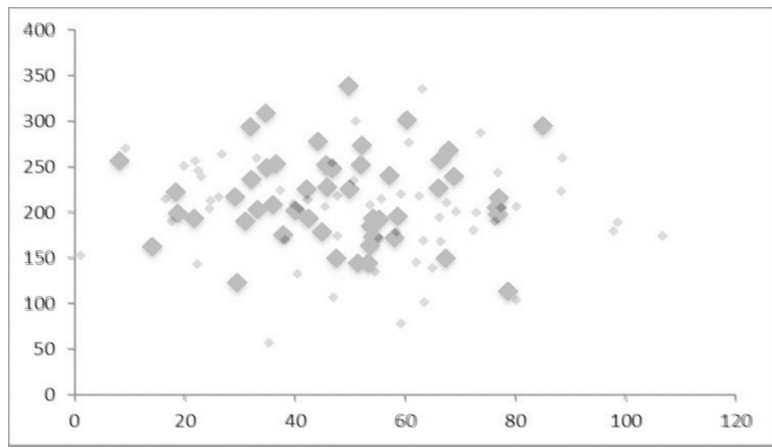

**Fig 1. Normal spatial distribution of cities (Pareto and lognormal tails).**

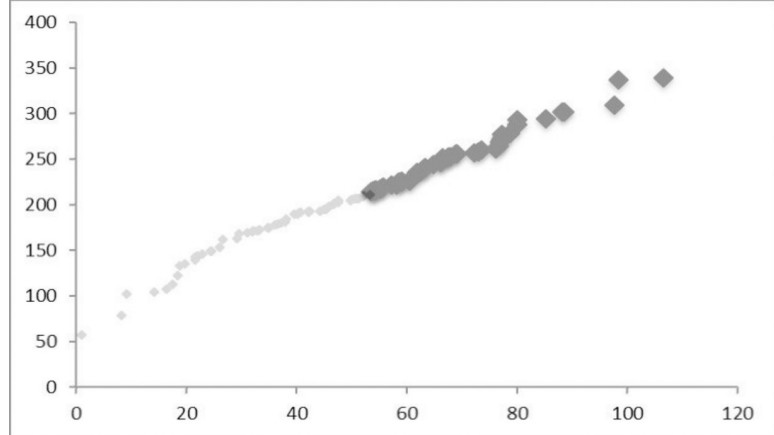

**Fig 2. Spatial distribution of cities with ranked coordinates (Pareto and lognormal tails).**

The two simulated cases are then compared with respect to the stability of $\alpha$ and the significance of the spatial parameters $\rho$ or $\lambda$ respectively. I hypothesise that with a normally distributed arrangement of coordinates the spatial weight parameters are insignificant and meaningless, representing a Zipf distribution of the upper tail similar to that of words in a language. However, when modifying the coordinates and the spatial distribution of cities by building clusters of cities with a different level of size we may expect some stronger and significant distance impact. The interesting question is then how stable the original Zipf distribution of the upper tail remains.

Finally, and in addition to the simulation analysis, this artificial exercise needs to be examined in the real world. The two questions are: do we find countries with significant spatial dependence in the rank-size distribution of cities and, if yes, how strong could it be? For that purpose I examine the spatial distance influence on Zipf's law with population data on US, German, British and Slovene urban areas respectively. For Slovenia as a particular case of young and small country, also natural cities (extracted from nocturnal satellite imagery) are explored in order to better capture the true functional urban space in that country. The respective image segmentation methodology is further explained in Bergs [33]. The source of data is the National Oceanic and Atmospheric Administration (NOAA) [34].

For the USA, the UK and Germany the database is truncated below 100,000 inhabitants. This is in line with Gabaix [3], Giesen and Südekum [35] and Brakman *et al.* [5]. Slovenia, being a former minor province of Yugoslavia, represents a lower scaling level with the primate city slightly more than twice as large as the above truncation point. Therefore cities larger than 10,000 inhabitants are covered. To demonstrate that all cities regarded are within the upper Pareto tail, a Shapiro-Wilk test was carried out for the log-transformed observations of population (or natural size).

As for the simulations, the SAR and the SEM model were used. In order to control for possible spatial dependence of the predictor in the country studies, SLX as an alternative spatial model was also tested. In all country models, the dependent variable was modified to $R$-1/2 (Gabaix-Ibragimov estimate) to avoid a potential bias of standard errors [36].

## 5 Results

First I take a look at the different results of the simulation exercise. Table 1 shows the full rank-size distribution, while Table 2 displays the estimates only for the upper (Pareto) tail.

Regarding case I, Zipf's law is only confirmed for the curtailed distribution: $\alpha \approx 1$ (Table 2). The lognormal extension with the smaller cities reduces the estimate of $\alpha$ to a large extent. As expected, the spatial coefficients $\lambda$ of the full distribution and $\rho$ of the Pareto tail are not significant. The normal distribution of coordinates of big and small cities leads to zero spatial dependence. With or without the spatial extension of the model the $\alpha$ coefficient remains the same. Zipf's law is well confirmed for the Pareto tail.

Case II shows the same rank-size distribution but ordered geographically along the coordinates. Now, there is a significant spatial error influence confirmed by $\lambda$ for the full distribution and a spillover effect $\rho$ for the Pareto tail. The absolute value of $\alpha$ decreases substantially when incorporating spatial dependence. Zipf's law cannot be further confirmed for the upper (Pareto) tail, even though the random-generated distribution had been a Pareto (3,1) one. At a first glimpse this finding may be puzzling, but it simply confirms the potential effect of a spatial arrangement with extremely enhanced spatial autocorrelation.

The simulated distributions of coordinates show that, in theory, spatial distance may have a potential impact on the coefficient of the rank-size distribution of cities. The stylised models above are however artificial and not likely to be encountered in the real world. Therefore data of four countries (three big ones, one small) are used to see how the spatial arrangement of cities may influence the estimate of rank-size distribution in the real world. As expected, the results generated are less spectacular than for simulation II but still suggesting spatial dependence to play a role in the rank-size distribution of cities for some countries:

For all samples the Shapiro-Wilk test rejects the Null hypothesis of normal distribution of the log-transformed observations in the upper tail, so the distributions regarded represent the respective Pareto tail.

Estimates for the rank-size distribution of US urban zones are very much in line with Zipf's law ($\alpha = 1.005$). The spatial error effect is small, however highly significant, and slightly improves the estimate. Hence, there is a very minor distance effect. A similar result is obtained for Germany: In the distribution of urban zones a significant spatial error improved the Pareto coefficient from 0.930 to 0.948. I also compared this estimate with German cities proper. In this sample, a coefficient of $\alpha = 1.239$ could not confirm Zipf's law; however even here a significant spatial lag effect moves the estimate slightly closer to Zipf ($\alpha = 1.227$).

In contrast to the USA and Germany, distance effects are insignificant in the case of the United Kingdom, both, for spatial error (not displayed) as well as spatial lag.

The Slovene case constitutes itself a bit different. There is no distance effect on the rank-size distribution of municipalities larger than 10,000 inhabitants. However, when viewing natural cities extracted from night satellite images we find that spatial spillovers ($\rho$) are significant at the $p<0.05$ level; the $\alpha$ coefficient, however, changes from 0.983 (within the Zipf tolerance of $\alpha = 1 \pm 0.1$) to 0.860. In this case, spatial dependence implies a deterioration of the rank-size distribution.

To complete the econometric findings by viewing a possible spatial lag of the predictor, the SLX model did not reveal significant spillover effects for any of the countries (Table 3). Hence, spatial dependence is only found for the dependent variable $R$.

Now, an interesting question is, from where such spatial dependence effects of rank may originate. For this purpose the local Moran I coefficients (LISA indicator) may offer insight [37]. The spatial weight matrices are those generated for the SEM/SAR regressions (data sources: see Table 4). In Fig 3 the resulting coefficients and the p-values for the USA, Germany, the UK and Slovenia are displayed and compared. The p-values are particularly important for the interpretation.

A striking evidence is that most of the 25 biggest urban zones in the USA exhibit significant spatial autocorrelation. Negative LISA coefficients prevail but there is a remarkable

**Table 3. Detection of spatial dependence by SLX regressions for selected countries (rank-size distribution of upper tails).**

| Endogenous variable: | USA (urb. zones) | Germany (urb. zones) | Germany (cities proper) | UK (urban zones) | Slovenia (municipalities) |
|---|---|---|---|---|---|
| ln(Rank-0.5) | | | | | |
| ln(Size) | -1.007 | -0.946 | -1.239 | -1.055 | -0.851 |
| (Standard error) | (0.004)*** | (0.020)*** | (0.140)*** | (0.013)*** | (0.073)*** |
| Constant | 17.234 | 15.162 | 18.077 | 15.966 | 9.592 |
| (Standard error) | (0.126)*** | (0.444)*** | (0.675)*** | (0.350)*** | (1.836)*** |
| θ | 0.002 | -0.037 | 0.107 | 0.008 | 0.034 |
| (Standard error) | (0.009) | (0.027) | (0.051) | (0.232) | (0.132) |
| Adj. $R^2$ | 0.993 | 0.978 | 0.988 | 0.989 | 0.943 |
| Breusch-Pagan test (p>$\chi^2$) | (0.000) | (0.000) | (0.000) | (0.000) | (0.678) |
| Obs. | 409 | 50 | 91 | 74 | 12 |

Note: The estimation of spatial dependence in the SLX model is limited to the spatial lag of the predictor variable. For Slovenia it was not possible to run this regression for the segmented VIIRS patches because coordinates are generated by image analysis (ImageJ). These have an equal area projection but are not transformable into kilometer distances by the Vincenty formula.

Data sources: See Table 4.

spread especially for the largest observations, e.g. a LISA coefficient of +3.9 for New York, outside the range of the Y axis. For Germany, the nine biggest urban zones exhibit significant spatial autocorrelation; the strongest being the Ruhr area on rank 1 (for German cities proper only five out of the biggest). For the UK, a significant LISA coefficient is only found for the first three urban zones. For Slovenia, spatial autocorrelation can only be established for the capital city area (Ljubljana). Getting back to the spatial econometric estimates, it is to be remembered that the most significant distance impact is found for the US urban areas, followed by German urban areas. This seems to be reflected by the LISA coefficients.

To summarize, the estimations discussed above display partly significant though modest spatial dependence in the city rank-size distribution of few selected countries. The existence of a significant type II error thus confirms the existence of spatial dependence. Probably there may be stronger or weaker such disturbances in other countries which are not regarded in this small sample. A large comparative study covering all countries, ideally over time and with more realistic weight matrices in the spatial econometric models (see earlier), could shed light on the global variation of spatial dependence in Zipf's law for cities.

## 6 Conclusion and further interpretation

Compared to Zipf's law for words in languages the results suggest that in case of cities, their spatial arrangement matters: Zipf's law for cities will behave like Zipf's law for words only if small and big cities are normally distributed in space. This is shown by the two simulations. In the predominant theory, during time cities may change their size, but the slope of the rank-size distribution remains rather stable [3]. This has been explained by its scale invariance and city growth independent of city size (Gibrat's law). Hence, the change of city ranks might be well explained by economic forces, but it is not directly visible in a changing slope of the rank-size distribution of cities. For this thread of argumentation spatial impact has no particular relevance. But studies combining Zipf's law with the Central Place theory show that a spatial relationship between centers of different layers is also in line with scale invariance [26–28]. Zipf's law can be also established in a model where distance exclusively governs the probability of city formation and growth

**Table 4. Detection of spatial dependence by SEM/SAR regressions for selected countries (rank-size distribution of upper tails).**

| Endogenous variable: ln(Rank-0.5) | USA (urban zones>100,000 inhabitants) | Germany (urban zones >100,000 inhabitants) | Germany (cities proper >100,000 inhabitants) | United Kingdom (urban zones>100,000 inhabitants) | Slovenia (municipalities >10,000 inhabitants) | Slovenia (segmented VIIRS patches; upper tail) |
|---|---|---|---|---|---|---|
| ln(Size) | -1.004 | -0.948 | -1.227 | -1.059 | -0.875 | -0.860 |
| (Standard error) | (0.004)*** | (0.012)*** | (0.014)*** | (0.013)*** | (0.057)*** | (0.071)*** |
| Constant | 17.171 | 14.735 | 18.738 | 15.801 | 10.275 | 5.765 |
| (Standard error) | (0.048)*** | (0.148)*** | (0.286)*** | (0.307)*** | (0.483)*** | (0.227)*** |
| λ | -0.781 | -3.520 | - | - | - | - |
| (Standard error) | (0.263)*** | (0.856)*** | - | - | - | - |
| ρ | - | - | -0.224 | 0.110 | -0.045 | -0.549 |
| (Standard error) | - | - | (0.100)* | (0.101) | (0.237) | (0.240)* |
| Log Likelihood | 499.317 | 57.624 | 86.480 | 73.348 | 6.395 | 8.389 |
| Wald-Test λ and ρ = 0 | | | | | | |
| χ2 | 8.412 | 16.880 | 4.981 | 1.177 | 0.036 | 5.255 |
| z-score: Moran's I (resid.) | -2.637 | -2.137 | - | - | - | - |
| (p) | (1.992) | (1.967) | - | - | - | - |
| Lagrange Multiplier (LM) | 7.357 | 3.872 | 4.099 | 1.438 | 0.048 | 3.465 |
| (p) | (0.007) | (0.049) | (0.043) | (0.230) | (0.827) | (0.063) |
| Robust LM | 7.034 | 4.902 | 4.487 | 1.121 | 0.065 | 3.502 |
| (p) | (0.008) | (0.027) | (0.034) | (0.290) | (0.798) | (0.061) |
| Obs. | 451 | 73 | 91 | 79 | 16 | 11 |
| OLS: ln(Rank-0.5) | | | | | | |
| ln(Size) | -1.005 | -0.930 | -1.239 | -1.056 | -0.881 | -0.983 |
| (Standard error) | (0.004)*** | (0.015)*** | (0.014)*** | (0.013)*** | (0.051)*** | (0.064)*** |
| Constant | 17.178 | 14.503 | 18.213 | 16.087 | 10.271 | 5.786 |
| (Standard error) | (0.049)*** | (0.186)*** | (0.169)*** | '(0.160)*** | (0.517)*** | (0.309)*** |
| Adj. R² | 0.993 | 0.982 | 0.989 | 0.989 | 0.955 | 0.959 |
| Shapiro-Wilk (p>z) | 0.000 | 0.000 | 0.000 | 0.000 | 0.001 | 0.000 |

Note: The choice between either SAR or SEM was determined by the z-score of Moran's I of the residuals and the Lagrange multiplier test statistic.

Data sources: https://simplemaps.com/data/us-cities for US urban zones (2019); https://www.destatis.de/DE/Themen/Laender-Regionen/Regionales/Gemeindeverzeichnis/Administrativ/Archiv/GVAuszugQ/AuszugGV2QAktuell.html for German cities proper (2020); https://www.citypopulation.de/en/germany/urbanareas/ for German urban areas (2019); https://www.citypopulation.de/en/uk/cities/ua/ for UK urban areas (2019); https://worldpopulationreview.com/countries/cities/slovenia for Slovene municipalities (2020); https://ngdc.noaa.gov/eog/viirs/download_dnb_composites.html for Slovene natural cities (October 2017) detected by night satellite imagery.

Missing geographical coordinates are compiled from: https://worldpopulationreview.com/countries/cities and Wikipedia.

[29]. There is thus reason to assume that dispersion and concentration forces determine the geographical distribution and centrality levels of cities, occasionally with more or less spatial dependence in their rank-size distribution. A spatial econometric approach suggests to shed light on such residual spatial dependence. If Gan *et al.* [2] were right, and Zipf's law represents nothing more than a pure statistical relationship, the extension of the model with spatial distance effects would not change $\alpha$. Where such spatial impact is significant, whether strong or modest, Zipf's law for cities is certainly more than a pure statistical phenomenon.

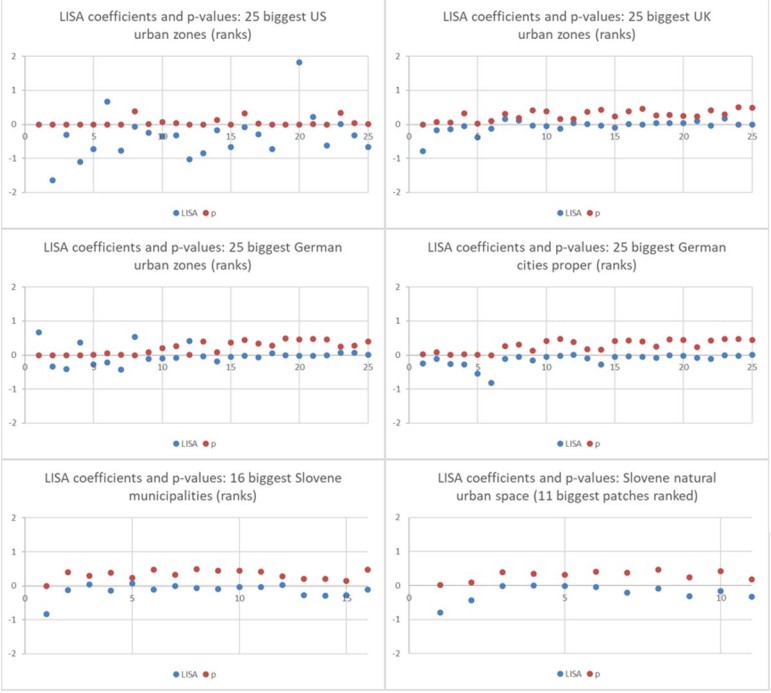

**Fig 3. LISA coefficients (local spatial autocorrelation) for the biggest urban areas.**

## Supporting information

**S1 File.**
(ZIP)

## Author Contributions

**Conceptualization:** Rolf Bergs.

**Data curation:** Rolf Bergs.

**Formal analysis:** Rolf Bergs.

**Funding acquisition:** Rolf Bergs.

**Investigation:** Rolf Bergs.

**Methodology:** Rolf Bergs.

**Project administration:** Rolf Bergs.

**Resources:** Rolf Bergs.

**Software:** Rolf Bergs.

**Supervision:** Rolf Bergs.

**Validation:** Rolf Bergs.

**Visualization:** Rolf Bergs.

**Writing – original draft:** Rolf Bergs.

**Writing – review & editing:** Rolf Bergs.

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
