## [Decision Letter · Decision Letter 0]

4 Aug 2020

PONE-D-20-16834

Spatial dependence in the rank-size distribution of cities

PLOS ONE

Dear Dr. Bergs,

Thank you for submitting your manuscript to PLOS ONE. After careful consideration, we feel that it has merit but does not fully meet PLOS ONE’s publication criteria as it currently stands. Therefore, we invite you to submit a revised version of the manuscript that addresses the points raised during the review process.

Please make your best effort to address both referees' concerns who are top specialists on the topic. This is very important for the process, since one of them recommends rejection and the other major revision. If you cannot address their concerns adequately, please explain in a separate note why this is so. In particular, if you disagree with them, please explain at length.

We look forward to receiving your revised manuscript.

Kind regards,

Yannis Ioannides

Academic Editor

PLOS ONE

Journal Requirements:

2.Thank you for stating the following in the Financial Disclosure section:

[Funding was provided by the European Commission via its Horizon2020 research funding (https://cordis.europa.eu/project/id/727988/de). The funders had no role in study design, data collection and analysis, decision to publish, or preparation of the manuscript.].   

We note that one or more of the authors are employed by a commercial company: PRAC

Reviewers' comments:

Reviewer's Responses to Questions

**Comments to the Author**

1. Is the manuscript technically sound, and do the data support the conclusions?

Reviewer #1: Yes

Reviewer #2: Partly

2. Has the statistical analysis been performed appropriately and rigorously? 

Reviewer #1: Yes

Reviewer #2: Yes

3. Have the authors made all data underlying the findings in their manuscript fully available?

Reviewer #1: Yes

Reviewer #2: No

4. Is the manuscript presented in an intelligible fashion and written in standard English?

Reviewer #1: Yes

Reviewer #2: Yes

5. Review Comments to the Author

Reviewer #1: See report.

Reviewer #2: This paper presents estimates from rank-size regressions that control for spatial effects. The main focus is on whether controlling for spatial effects influences estimates of power law exponents significantly. In the Netherlands (2011), Slovenia (2017) and Austria (2017) my reading of the results suggests the influence is weak and that the estimates of spatial dependence suggest moderate to weak dependence.

Regarding methodology:

1. We are left wondering why focus on Netherlands, Slovenia and Austria rather than more widely studied countries like the US or even better a very comprehensive list of countries, ideally at several points in time. The more countries the better in my opinion, but if a selection is made the basis of that selection needs to be explained. Any statistical methodology is undermined if it is applied to an arbitrary subset of the potential data.

2. I would like to see the results from simple rank-size regressions alongside the regressions that control for spatial dependence.

3. Would it be econometrically sensible to control for spatial dependence in Gabaix-Ibragimov regressions like those of Table 3? If you cannot answer this question, it may be interesting to nevertheless run these regressions subject to appropriate disclaimers in order to allow direct comparisons of estimates.

4. I would prefer to see the discussion surrounding simulated data significantly condensed as I am not sure it adds much.

5. It may be worth noting that combinations of lognormal-power laws similar to those used in the simulations have been studied by Ioannides & Skouras 2013.

Regarding data: No specific link to data is provided so I am not sure whether the source provided in the last sentence of section 2 is sufficient. I would prefer to see a link to the data actually used in the regressions (including to simulated data), not a link e.g. to a NOAA source from which the data were derived after extensive manipulation according to methods published elsewhere.

6. PLOS authors have the option to publish the peer review history of their article (what does this mean?). If published, this will include your full peer review and any attached files.

Reviewer #1: No

Reviewer #2: No

---

## [Author Response · Author response to Decision Letter 0]

15 Sep 2020

Re.: Spatial dependence in the rank-size distribution of cities

Rebuttal letter #20-16834

Dear editor and reviewers,

Thank you very much for your highly valuable critique and suggestions. In the revised manuscript I tried my best to address all your points raised. Originally, my paper had been a secondary statistical outcome of our ongoing research project on rural-urban interaction in the EU. I submitted the paper in May 2020 to the preprint server arXiv. Shortly afterwards I was kindly invited by PlosOne to submit my/our research to a forthcoming special issue on „Complex city systems“. In short, this is the background of my original paper. Inspired by your points raised I have now completely revised the manuscript, also using larger datasets for a better empirical foundation. I thus hope that the effort has helped to meet the expectations of the reviewers and the editor. 

Reviewer 1:

1.There is really no economic model here; the model is purely statistical. The gravity model is not derived in this context from primitives, and is rather hokey. This is, in a way, a step backward.

I fully agree with this view. It is not sufficient to determine a finding purely statistically when arguing from the viewpoint of economics. Zipf’s law is however extremely tricky when it comes to explaining it with economic theory (Krugman labeled the law as „spooky“). I now tried to solve the issue of the insufficiently explained relationship with gravity in a new section by first going back to Zipf’s own explanation of the „principle of least effort“ (lines 73-113) and then to link this very early insight (Zipf’s lemma) to the respective recent findings in the context of Central Places theory (with a reference to the suggested paper of Mori, Smith and Hsu and a second paper on that by Jiang). Both papers make clear that distance and spatial dependence matters in these Zipf-style regressions, so that they offer important justification to consider a spatial econometric approach (lines 119-138).

2.The pre-Gabaix and post-Gabaix periods are distinguished by the presence of economic models in the post-Gabaix work. The problem with these models is often that they are designed to generate Zipf-like laws, and nothing else.

This was a useful hint for me. My understanding is that Gabaix‘ model of normalized zero growth of cities determined by the underlying Gibrat’s law is only part of explaining scale invariance in Zipf’s law. The papers by Mori et al. (2020) and Jiang (2020) show that the functional differentiation of centrality matters, also with explanatory power for the fractal dimension of the law. But there seems even no contradiction between size independent (random) growth and a deterministic evolution of urban space. Therefore, for my argumentation I also added the important findings from Rybski‘s et al. paper on distance-weighted city growth that confirms scale invariance and Zipf’s law with „spatial explicit preferenial attachment“ (lines 139-159).

3.This is not the first paper to notice the transition between Pareto at the top of the distribution and lognormal at the bottom (lines 55-69); Ioannides and Skouras, JUE, 2013. Not even cited.

It would be a misunderstanding to interpret my formulations as a claim to be the first one noticing this peculiarity.This is not al all the case. For my earlier paper (Rev Reg Res 2018 38(2)) I had explored the well-known controversy between Malevergne et al. and Eeckhout. So, this knowledge obtained I just took over from that earlier research. Reading the paper by Ioannides and Skouras (2013) has much further enriched the argumentation to differentiate the distribution form of city rank-size.The JUE paper is now addressed and cited (lines 54-64).

4. There is a theory for why distance should matter in these Zipf-style regressions, called Central Place theory, which is almost as famous as Zipf’s law. It is the long-time pursuit of Wen-Tai Hsu. See for a recent example, Mori, Smith and Hsu, PNAS, 2020, 117(12) 6469-6475.

I fully agree. See my explanations on point 1 above

5. Given the spatial dependence in rank R (the dependent variable) I wonder if there is also spatial dependence in population S, the independent variable.

For the four country studies I now additionally tested spatial dependence of the predictor and the dependent variable in a Spatial Durbin model. This procedure did not properly work as estimates are mostly outside their allowed intervals. A simpler approach to address spatial dependence of the predictor is the Spatial-lag-of-X (SLX) model that directly shows spatial spillovers of the variable S by OLS estimation. In contrast to the SEM/SAR models, no significant spatial dependence was found for S. This finding is now documented by an additional table (lines 311-315) and two further paragraphs inserted (lines 211-217 and 346-348).

6. The structure of spatial dependence in the regression should really be derived from the underlying economics

This critique applies to every spatial econometric modelling that defines W simply as a contiguity or inverse distance matrix. Therefore, on the one hand, inverse distance that I used in my Stata exercises is admittedly not more than a proxy. A more realistic matrix reflecting the effort (real costs) to cover distance, given the heterogeneity of space, such as local topography or energy resources available, would be in fact a superior solution. However, the bottleneck for that is missing data. I have therefore added a clarifying paragraph stressing this general weakness (lines 218-227). On the other hand, a spatial econometric approach would shed light on the endogeneity of spatial distance in the Zipf regressions. This is a specific advantage, because here peculiar forms of deterministic spatial evolution (e.g. the German Ruhr area) suggest spatial dependence in the Zipf regressions already by viewing a map (clusters of big cities with often minor centrality relevance). In those cases, spatial dependence does not necessarily conceal a distribution that is not in line with Zipf (i.e. a false positive signal). On the contrary, as estimates for the US and Germany show, spatial dependence can also lead to estimates closer to Zipf’s law. Therefore, I additionally had a look at the city distribution in the upper tails (25 biggest cities for the USA, Germany and the UK, the 16 biggest for Slovenia) in terms of local spatial autocorrelation (lines 355-362). For the US and Germany there are many significant local Moran-I (LISA) coefficients, while this is not the case for the UK and Slovenia. Results of the local Moran I analysis thus correspond to the SEM/SAR estimates. A further figure on the LISA analysis was added to the paper (lines 353-354). 

Reviewer 2:

1. We are left wondering why focus on Netherlands, Slovenia and Austria rather than more widely studied countries like the US or even better a very comprehensive list of countries, ideally at several points in time. The more countries the better in my opinion, but if a selection is made the basis of that selection needs to be explained. Any statistical methodology is undermined if it is applied to an arbitrary subset of the potential data.

I fully understand and agree to this concern. In fact, the research is funded by the European Commission in the context of a project dealing with rural-urban interaction covered by some EU country studies including Austria, Slovenia and the Netherlands where we had tested spatial size (segmented by cluster analysis of night satellite images) with Zipf’s law. I just used those data for the first manuscript. In addition to that, the EU funding of an open access paper should also be justified by the respective project context, e.g. the countries covered by that project. Hence, the selection had not been arbitrary. But still, the reviewer’s argument is admittedly too strong. Therefore, an alternative justification could be derived from the theoretical context elaborated, and hopefully this is convincing. I therefore explored databases on urban areas for three bigger countries (USA, UK and Germany) and left only Slovenia as a particular case of a young EU country (former province of Yugoslavia) for comparison purposes (Results for the Netherlands and Austria are now removed). It was, however, not possible for the three bigger countries to extract and transform the VIIRS night-light data in order to classify natural urban space. There are millions of observations in the highly resolved digital images for which outliers in such skewed pixel distributions need first to be removed by a quite demanding procedure. The computing capacity in our small institute is not sufficient for procedures with such extremely large datasets. Therefore, for the Zipf style regressions on the US, the UK and Germany I used population data as usual. Here I deemed it useful not to take cities proper but (functional) urban areas, because evidence has shown that urban areas much better represent a Zipfian distribution. A further comparative analysis over time was not possible given the limited time for the revision. But the original purpose of the paper was to show that there can be spatial dependence in Zipf’s law for cities rather than exploring change of spatial dependence over time in Zipf’s law. Nevertheless, such an extended comparative or panel approach combined with a more realistic spatial weight matrix (cf. Point 6 of reviewer 1) is now discussed as an interesting open question for future research on this thread of regional science (lines 224-227).

2. I would like to see the results from simple rank-size regressions alongside the regressions that control for spatial dependence.

These simple rank-size regressions had been (and still are) displayed in the tables alongside the spatial regressions including their interpretation. Those might have been either overlooked when reading the manuscript, or I have misunderstood the point raised. Nevertheless, I fully agree that without the simple rank-size regressions the respective spatial regressions would hardly be meaningful (see tables 1, 2 and 3: rows „OLS ln(Size)“ and the respective remarks in the section „Results“) .

3. Would it be econometrically sensible to control for spatial dependence in Gabaix-Ibragimov regressions like those of table 3? If you cannot answer this question, it may be interesting to nevertheless run these regressions subject to appropriate disclaimers in order to allow direct comparisons of estimates.

Yes, my earlier idea had been anyway to generally apply the Gabaix-Ibragimov approach for the country estimates, i.e. also for the ML estimator that is needed in the SEM/SAR regressions. This approach had been also pursued by le Gallo & Chasco (2008) in one of the very few spatial econometric studies on Zipf’s law. Estimates differ only minimally from the simple regression model. Now, all the country estimates displayed are Gabaix-Ibragimov ones (lines 271-272).

4. I would prefer to see the discussion surrounding simulated data significantly condensed as I am not sure it adds much.

I have deleted the simulations III and IV and just concentrated on the two potential extreme cases of normally distributed coordinates versus spatially ranked ones. This simulation should reveal the potential of spatial dependence in such Zipf regressions.

5. It may be worth noting that combinations of lognormal-power laws similar to those studied in the simulations heve been studied by Ioannides & Skouras 2013

As mentioned in the answer on point 3 of reviewer 1, the suggested JUE paper is now addressed and cited. 

6. No specific link to data is provided so I am not sure whether the source provided in the last sentence of section 2 is sufficient. I would prefer to see a link to the data actually used in the regressions (including to simulated data), not a link e.g. to a NOAA source from which the data were derived after extensive manipulation according to methods published elsewhere

This is absolutely right. However, the guidelines say that „PLOS journals require authors to make all data necessary to replicate their study’s findings publicly available without restriction at the time of publication.“ During the online submission procedure I was therefore uncertain whether to attach the data files together with the manuscript or only once such a paper is accepted for publication. Shortly afterwards I contacted the editorial staff of PlosOne asking whether I can still upload the data for the peer review process. I was told that it will be fine to provide the full datasets once the manuscript has been accepted, so there was no need for me to do anything by then. 

Reviewer 2 also remarked that estimates on Austrian, Slovene and Dutch natural urban space suggest minor to moderate spatial dependence. This is true, and I also believe that there is hardly any country where spatial dependence will change the estimate of the rank-size distribution to a large extent. The purpose of my paper was to show that there can be minor but significant spatial dependence; this is also confirmed by the estimates for the USA and Germany. It was less my intention to uncover major disturbance induced by spatial dependence. But with statistically significant spatial dependence Zipf’s law would not be a purely tautological phenomenon of a pre-determined spurious correlation as put forward by some authors.

---

## [Decision Letter · Decision Letter 1]

24 Dec 2020

PONE-D-20-16834R1

Spatial dependence in the rank-size distribution of cities

PLOS ONE

Dear Dr. Bergs,

Thank you for submitting your manuscript to PLOS ONE. After careful consideration, we feel that it has merit but does not fully meet PLOS ONE’s publication criteria as it currently stands. Therefore, we invite you to submit a revised version of the manuscript that addresses the points raised during the review process.

We look forward to receiving your revised manuscript.

Kind regards,

Yannis Ioannides

Academic Editor

PLOS ONE

Additional Editor Comments (if provided):

Dear author:

I have immense respect for both referees, and wish to encourage you to revise according to

Reviewer 2, who submitted a detailed report.

I also want you to heed the comments of Reviewer 1, who now is very encouraging in his direct communication with me. I agree with him that section 2 needs more work, so as the paper be more appealing to economists who read it. And, most certainly, this is a worthy goal. Reviewer 1 writes, inter alia:

" I think that the author has done as good a job as it is possible to

do to address my comments regarding the empirical part of the paper. But I find the theory

(Section 2) to be very annoying. It is not theory in a sense that a decent economist would

recognize, as it is in the tradition of econophysics rather than mainstream economics. The

econophysics models tend to be fairly mechanical models (including stochastic elements)

rather than using an equilibrium based on individual optimization. A hint is that prices

are nowhere to be found in this paper."

There is a well-developed theory with behavioral foundations, including notably Gabaix's QJE paper, which you cite, and material in Ch. 8 of Yannis M. Ioannides, From Neighborhoods to Nations, Princeton University Press, 2013. I think heeding Reviewer 2's critique will improve the paper enormously.

All the best!

Looking forward to reviewer an updated version, which I very much hope that you will undertake.

Yannis M. Ioannides

Academic Editor

Reviewers' comments:

Reviewer's Responses to Questions

**Comments to the Author**

1. If the authors have adequately addressed your comments raised in a previous round of review and you feel that this manuscript is now acceptable for publication, you may indicate that here to bypass the “Comments to the Author” section, enter your conflict of interest statement in the “Confidential to Editor” section, and submit your "Accept" recommendation.

Reviewer #1: All comments have been addressed

Reviewer #2: (No Response)

2. Is the manuscript technically sound, and do the data support the conclusions?

Reviewer #1: Yes

Reviewer #2: Partly

3. Has the statistical analysis been performed appropriately and rigorously? 

Reviewer #1: Yes

Reviewer #2: Yes

4. Have the authors made all data underlying the findings in their manuscript fully available?

Reviewer #1: Yes

Reviewer #2: Yes

5. Is the manuscript presented in an intelligible fashion and written in standard English?

Reviewer #1: Yes

Reviewer #2: Yes

6. Review Comments to the Author

Reviewer #1: (No Response)

Reviewer #2: This revision has addressed all the concerns I raised in my first report and the author has clearly made a serious effort at improving the quality of his paper.

However, the revision has also revealed some new problems which I summarize below:

1. My reading of the new empirical results is that the author is able to detect only a very weak impact of distance between cities on power exponent estimates in the four countries he examines. I think the author should say this more clearly (instead in the abstract, he states his finding as "distance matters"). The author should also avoid conflating distance with "spatial effects" - there may well be other spatial effects he has not tested for. There may also be distance effects in other countries or data, so the author should be more explicit that he is conducting an analysis with limited power and subject to significant type 2 error when interpreted as a test of "spatial dependence in the rank-size distribution of cities" (perhaps the title itself should be modified to reflect the more modest nature of the analysis). Summarizing, there is a little too much overselling for my taste, but this may be a style issue.

2. The results of the second simulated data set in Table 2.II are puzzling. They suggest that even when the data really is generated to satisfy Zipf's law, the econometric approach used reveals a significant deviation from Zipf's law. This suggests a problem with the econometric method or its application or the data. Maybe I am missing something, but if so the author needs to explain.

3. I am not fully comfortable with the author's description of the data used in the simulation. The author says he draws the "upper" 50 cities from a Pareto and the "lower" 59 from a lognormal. If the draws are really random, the largest lognormal draw could be larger than the smallest Pareto draw, but the phrasing suggests this cannot happen, or at least did not happen in the two draws the author used. The author should explain this more clearly and make sure he isn't choosing a sample with the properties he wants. While I don't expect the author to do this at this stage, the proper way to simulate this data would be from a single distribution which had both a lognormal and a pareto component.

4. It should be made clearer that the value of the simulations is to demonstrate that the chosen econometric methodology is powerful enough to detect distance effects if the patterns are sufficiently strong. In my view the simulations are purely a prelude to motivate the empirical analysis.

5. I would like to see more detailed table legends, so that tables can be interpreted without having to refer to the text. We have to guess what Columns I and II mean. Please explain each item in the table in detail in the legend - a little spoon feeding for the reader can only help.

7. PLOS authors have the option to publish the peer review history of their article (what does this mean?). If published, this will include your full peer review and any attached files.

Reviewer #1: No

Reviewer #2: No

---

## [Author Response · Author response to Decision Letter 1]

22 Jan 2021

Re.: Spatial dependence in the rank-size distribution of cities

Rebuttal letter #20-16834

Dear editor and reviewers,

thank you again for your further effort to review the revised version of my paper. The review reports are very encouraging. All suggestions are highly valuable and could be addressed in the revised manuscript. I hope this revised manuscript meets your expectations and look forward to your response. Below you find my answers separately on all comments. 

Reviewer #1 (including the suggestion of the editor):

" I think that the author has done as good a job as it is possible to do to address my comments regarding the empirical part of the paper. But I find the theory (Section 2) to be very annoying. It is not theory in a sense that a decent economist would recognize, as it is in the tradition of econophysics rather than mainstream economics. The econophysics models tend to be fairly mechanical models (including stochastic elements) rather than using an equilibrium based on individual optimization. A hint is that prices are nowhere to be found in this paper."

There is a well-developed theory with behavioral foundations, including notably Gabaix's QJE paper, which you cite, and material in Ch. 8 of Yannis M. Ioannides, From Neighborhoods to Nations, Princeton University Press, 2013. I think heeding Reviewer 2's critique will improve the paper enormously

Reply: I have now substantially revised and complemented section 2. Zipf’s law explained with early considerations by Zipf himself is now complemented by considerations of the economics of agglomeration. A useful starting point seemed to me the indivisibility of space (Starret’s spatial impossibility theorem) as a precondition for an equilibrium that cannot be explained by Zipf’s lemma. Here I found it worth to resort to the evolution of urban economic modeling from the 1970s to new geographical economics since around 2000 to grasp the increasingly „spatial“ understanding of Zipf’s law from the economics viewpoint (in particular the more recent contributions of the Central Place theory). In addition to the paper of Gabaix (1999b) I now discussed the relevant findings of Fujita et al (1999), Brakman et al. (2009) and those in more recent books and papers stressing the importance of Central Place theory for the explanation of Zipf’s law. Here I resorted to the book of the editor (Ioannides 2013), Hsu (2013), Mori et al. (2020) and Jiang (2017) [LINES 118-130; 142-182]. A useful argument for taking the path of spatial econometrics I also found in Dobkins and Ioannides (2001) on spatial interactions among US cities. [LINES 210-215]. 

Reviewer #2: This revision has addressed all the concerns I raised in my first report and the author has clearly made a serious effort at improving the quality of his paper.

However, the revision has also revealed some new problems which I summarize below:

1. My reading of the new empirical results is that the author is able to detect only a very weak impact of distance between cities on power exponent estimates in the four countries he examines. I think the author should say this more clearly (instead in the abstract, he states his finding as "distance matters"). The author should also avoid conflating distance with "spatial effects" - there may well be other spatial effects he has not tested for. There may also be distance effects in other countries or data, so the author should be more explicit that he is conducting an analysis with limited power and subject to significant type 2 error when interpreted as a test of "spatial dependence in the rank-size distribution of cities" (perhaps the title itself should be modified to reflect the more modest nature of the analysis). Summarizing, there is a little too much overselling for my taste, but this may be a style issue.

Reply: I agree with this comment. My original research interest was led by the idea that spatial dependence could potentially matter, and here the emphasis still had been on the simulations which reveal major potential of such disturbances. Now the emphasis is on the country studies where spatial autocorrelation detected is weak but still partly significant. Hence I changed the title of the paper and the wording in a number of paragraphs in particular to avoid suggesting something like globally valid relationships [Abstract; LINES 320-322; 444-449; 468-469]. I also revised the formulations conflating spatial effects with distance effects [Various revisions of wording in the text; cf. track-change file].

2. The results of the second simulated data set in Table 2.II are puzzling. They suggest that even when the data really is generated to satisfy Zipf's law, the econometric approach used reveals a significant deviation from Zipf's law. This suggests a problem with the econometric method or its application or the data. Maybe I am missing something, but if so the author needs to explain.

Reply: With this simulation I deliberately aimed to generate a most powerful disturbance effect on the Zipf coefficient by maximising spatial autocorrelation. In the former manuscript this was only commented by a half-sentence. In the new version I added a further explanation that is in fact central in a way as it shows the theoretical potential of such effects [LINES 396-398]. By the way, with a repeated estimation to rule-out error I got the same results. 

3. I am not fully comfortable with the author's description of the data used in the simulation. The author says he draws the "upper" 50 cities from a Pareto and the "lower" 59 from a lognormal. If the draws are really random, the largest lognormal draw could be larger than the smallest Pareto draw, but the phrasing suggests this cannot happen, or at least did not happen in the two draws the author used. The author should explain this more clearly and make sure he isn't choosing a sample with the properties he wants. While I don't expect the author to do this at this stage, the proper way to simulate this data would be from a single distribution which had both a lognormal and a pareto component.

Reply: I have now explained how this hybrid distribution was generated. My original idea had been to generate just one typical rank-size distribution of cities and as long as this can be sufficiently rigged by distorting the location of cities via the coordinates the exercise would prove the potential existence of spatial dependence in Zipf’s law. It is right that the lower (lognormal) tail could be also much larger. This would have an influence on the estimation of the entire distribution, however not when looking at the upper Pareto tail. This is now explained in section 4 [LINES 289-296].

4. It should be made clearer that the value of the simulations is to demonstrate that the chosen econometric methodology is powerful enough to detect distance effects if the patterns are sufficiently strong. In my view the simulations are purely a prelude to motivate the empirical analysis.

Reply: I fully agree with that. The issue is now clarified [LINES 68-70].

5. I would like to see more detailed table legends, so that tables can be interpreted without having to refer to the text. We have to guess what Columns I and II mean. Please explain each item in the table in detail in the legend - a little spoon feeding for the reader can only help.

Reply: I agree to that and added respective explanatory notes to Tables 1 and 2 [LINES 354-357; 368-371]. 

Further corrections

In addition to revisions suggested by the reviewers and the editor I also corrected the SEM and SAR estimates of the four countries and the respective LISA figures. For the population based estimations it was necessary to change the geographical projection of the raw data into an equal area one. The correction reveals only negligible differences, most of them even improving the estimates expected [LINES 374-375; 431-432]. In addition some few transcription errors were corrected. There was no need to change the projection for the simulations and the estimation on segmented natural cities in Slovenia. In those cases, coordinates already represent an equal area projection. There was also no need to modify the projection of the SLX estimates as the respective Stata command automatically calculates the true distances by the Vincenty formula. For Slovene municipalities (>10,000 inhabitants) the SLX estimation was to be corrected because of a wrong entry of the distance threshold. But this has not changed the insignificant result.

Some few further style and orthographic corrections can be identified by inspecting the track-change version of the manuscript.

---

## [Editor Report · Decision Letter 2]

27 Jan 2021

Spatial dependence in the rank-size distribution of cities - weak but not negligible

PONE-D-20-16834R2

Dear Dr. Bergs,

We’re pleased to inform you that your manuscript has been judged scientifically suitable for publication and will be formally accepted for publication once it meets all outstanding technical requirements.

Kind regards,

Yannis Ioannides

Academic Editor

PLOS ONE

Additional Editor Comments (optional):

Dear author:

thank you for patiently and diligently dealing with the editorial comments on your submission.

I am happily recommending acceptance of your submission for publication by PLOS One.

All the best

Yannis M. Ioannides

Academic Editor
---

## [Editor Report · Acceptance letter]

29 Jan 2021

PONE-D-20-16834R2 

Spatial dependence in the rank-size distribution of cities –  weak but not negligible 

Dear Dr. Bergs:

I'm pleased to inform you that your manuscript has been deemed suitable for publication in PLOS ONE. Congratulations! Your manuscript is now with our production department. 

Kind regards, 

on behalf of

Dr. Yannis Ioannides 

Academic Editor

PLOS ONE